# Cutaneous Melanoma and Glioblastoma Multiforme Association—Case Presentation and Literature Review

**DOI:** 10.3390/diagnostics13061046

**Published:** 2023-03-09

**Authors:** Olguța Anca Orzan, Călin Giurcăneanu, Bogdan Dima, Monica Beatrice Dima, Ana Ion, Beatrice Bălăceanu, Cornelia Nițipir, Irina Tudose, Cătălina Andreea Nicolae, Alexandra Maria Dorobanțu

**Affiliations:** 1Dermatology Department, “Carol Davila” University of Medicine and Pharmacy, 020021 Bucharest, Romania; 2Dermatology Department, “Elias” University Emergency Hospital, 011461 Bucharest, Romania; 3Oncology Department, “Carol Davila” University of Medicine and Pharmacy, 020021 Bucharest, Romania; 4Oncology Department, “Elias” University Emergency Hospital, 011461 Bucharest, Romania; 5Anatomic Pathology Laboratory, “Elias” University Emergency Hospital, 011461 Bucharest, Romania

**Keywords:** glioblastoma, melanoma, cancers, astrocytomas, nervous system tumors, melanoma–astrocytoma syndrome, *CDKN2A*

## Abstract

The occurrence of both melanoma and glioma was first suggested by the observation of a familial association between these conditions, which was later confirmed by the description of the melanoma–astrocytoma syndrome, an extremely rare, inherited affliction in which people have an increased risk of developing melanoma and nervous system tumors. Taking into consideration the common embryologic precursor, the neuroectoderm, it was hypothesized that this syndrome is associated with a genetic disorder. While some families with germline *CDKN2A* mutations are prone to develop just melanomas, others develop both melanomas and astrocytomas or even other nervous-system neoplasms. Herein, we report the case of a 63-year-old male patient with no personal or family history of malignancy who had primary melanoma followed by glioblastoma. Our case report suggests that the occurrence of both melanoma and glioblastoma is most likely not coincidental but instead linked to genetic mutations of common embryologic precursors or signaling pathways.

## 1. Introduction—Literature Review

Glioblastoma is a high-grade malignant glioma and is additionally one of the most harmful and lethal disorders of the central nervous system. Melanoma is classified as the most dangerous skin cancer with high mortality rates.

During embryogenesis, the skin is formed through a complex process involving cells derived from both ectodermal and endodermal germ layers. The former is further divided into surface ectoderm, which gives rise to epidermal keratinocytes, and neuroectoderm, which forms the neural tube through invagination (neurulation). Furthermore, the posterior portion of the neural tube separates into the neural crest which generates nervous cells, both neurons and glial cells. In addition, another cell type that derives from the neural crest is the melanocyte, which migrates towards the epidermis [1].

In 1993, Kaufman et al. described the melanoma–astrocytoma syndrome, reporting on eight members of a family spread over three generations that developed cutaneous malignant melanoma and/or cerebral astrocytoma [2]. Taking into consideration the common embryologic precursor, the neuroectoderm, it was hypothesized that this syndrome is associated with a genetic disorder [2].

One of the main candidates is *CDKN2A* (Cyclin-Dependent Kinase Inhibitor 2A), situated on chromosome 9p21, because it is involved in neuroblastoma cell growth, while also representing a high penetrance susceptibility gene for familial cutaneous melanoma and sporadic multiple cutaneous melanoma syndromes [3,4,5].

*CDKN2A* is a tumor suppressor gene with a unique characteristic of encoding two distinct proteins by alternative splicing [6]. The two transcripts share exons 2 and 3 but differ through the first exon: when exon 1α is used the resulting protein is p16^INK4A^, whereas exon 1β encodes p14^ARF^, by employing an alternative reading frame [6].

*CDK4* encodes a catalytic subunit of a Ser/Thr protein kinase that takes part in the G1 phase progression of the cell cycle through phosphorylation of the Retinoblastoma protein (Rb). p16^INK4A^ binds through arginine to *CDK4* and inhibits its function, thus arresting the cell cycle in the G1 phase, preventing aberrant cell growth. Cen et al. outline the importance of p16^INK4A^-*CDK4*-Rb pathway mutations in the development of glioblastoma multiforme, as well as in predicting the response to targeted therapies with cyclin-dependent kinase inhibitors [7].

On the other hand, p14^ARF^ stabilizes another tumor suppressor protein, p53, by protecting it from mediated degradation through E3 ubiquitin-protein ligase MDM2 (Murine Double Minute/Mouse Double Minute 2 homolog) [6]. Unlike p16^INK4A^, p14^ARF^ can arrest the cell cycle in both G1 and G2 phases [6]. 

Petronzelli et al. cite exon 1β mutations affecting p14^ARF^ as the main trigger for the development of nervous system tumors, while larger germline deletions of *CDKN2A*, most likely exon 2 defects that affect both p16^INK4A^ and p14^ARF^, are associated with melanoma-nervous system tumor syndromes [8]. Similar results have been published by Knappskog et al. who reported on a Norwegian melanoma-prone family (four out of fifteen members, two of them also having multiple cutaneous melanomas) which harbored a large *CDKN2A* deletion that led to truncated p14^ARF^ mRNA (lacking exon 2) and loss of p16^INK4A^ mRNA expression from the affected allele, thus altering the gene transcripts [9].

Loo et al., by sequencing exons and adjacent intronic regions, hypothesized that mutations affecting the noncoding, intronic regions of *CDKN2A* could also be responsible for altered p16^INK4A^ transcripts that are unable to properly bind to *CDK4* and inhibit it, thus explaining why some melanoma-prone families do not exhibit *CDKN2A* coding mutations [10].

It is well known that a positive family history is a significant risk factor for other family members to develop melanoma. Harland et al. identified in six English melanoma pedigrees a mutation situated deep in intron 2 of *CDKN2A*, more precisely at 105 bases from the 5′ end, towards exon 3, which results in aberrant splicing of mRNA [11]. The challenge in identifying mutations in noncoding regions is that they are extremely variable depending on the studied geographic population. Moreover, some mutations may not be included in the sequences amplified by RT-PCR due to the limitations of the primer selection.

*CDKN2A* gene silencing caused by mutations, deletions, or promoter hyper-methylation, has also been reported in a variety of other human cancer cell lines, such as: pancreatic adenocarcinoma, non-small cell lung cancer, Burkitt’s lymphoma, gastric cancer, colorectal cancer, prostate cancer, breast cancer, and ovarian cancer [12].

Goldstein analyzed 189 melanoma-prone families, 42 of them with a positive history of pancreatic cancer, and identified 67 different germline *CDKN2A* mutations, exon 2 being the most frequently mutated in both groups [13]. Additionally, missense mutations were the most prevalent (56%) for both melanoma and pancreatic cancer families [13].

Unlike *CDKN2A*, *MC1R* is a low-moderate-risk melanoma susceptibility gene, and it encodes the G-protein-coupled receptor for α-melanocyte stimulating hormone (α-MSH) [4]. When α-MSH binds to *MC1R* in the presence of UVB-exposure, it triggers the activation of adenylate cyclase which leads to an increase in cAMP production [4]. In turn, an elevated level of cAMP causes a switch in melanin production from red/yellow pheomelanin pigments to protective brown/black eumelanin pigments [4]. In addition, *MC1R* induces cAMP-dependent expression of catalase and superoxide dismutase which protect the UVB-exposed cells from oxidative stress, thus reducing the incidence of DNA defects [14].

Harding et al. studied the diversity in *MC1R* variants on different continents and discovered that in non-African populations there are many *MC1R* aminoacid variants which account for the large diversity of pigmentation patterns [15]. On the other hand, in Africa, *MC1R* variants were evolutionary eliminated due to a strong functional constraint of conversion to eumelanin pigments [15].

*MC1R* variants are incapable of switching the pigmentation pattern to eumelanin production and this explains why up to 75% of skin type I people carry a variant *MC1R* allele, whereas in darker skin types the percentage of *MC1R* variants vastly decreases [15]. Three main *MC1R* variants have been found to be associated with red hair and fair skin: *R151C* (*Arg151Cys*), *R160W* (*Arg160Trp*) and *D294H* (*Asp294His*), and have been grouped under the term of “red hair color” variants [16]. Moreover, *MC1R* “red hair color” variants are also correlated with other pigmentation characteristics, such as: freckling, propensity to burn and failure to tan after sun exposure [17]. All these characteristics are known risk factors for the development of cutaneous malignant melanoma [17]. Moreover, Palmer et al. calculated that *MC1R* variants are associated with a 2.2-fold risk increase of melanoma and that the effect is additive in individuals carrying more than one *MC1R* variant [17]. In addition, this predisposition also persists in individuals with dark skin, thus confirming that *MC1R* variants are independent risk factors for melanoma, apart from their influence in pigmentation patterns [4].

Box et al. studied 15 Australian melanoma pedigrees harboring *CDKN2A* mutations and discovered that families carrying *MC1R* “red hair color” variants had a *CDKN2A* mutation penetrance of 84%, whereas families with a normal *MC1R* gene presented a *CDKN2A* mutation penetrance of 50% [18]. Therefore, apart from being an independent risk factor for melanoma, *MC1R* variants also increase the penetrance of other high-susceptibility mutations (*CDKN2A*). In addition, when a *MC1R* variant was present, the mean age at onset of melanoma was also considerably lower [18]. 

Cao et al. reported that UVB-induced inactivation of *PTEN* (through phosphorylation of Ser380 and Thr382/383) and activation of Akt (through phosphorylation of Ser473) were significantly increased in cells expressing *MC1R* “red hair color” mutant variants, compared to cells expressing wild-type non-mutated *MC1R* (*WT-MC1R*) [19]. In addition, *WT-MC1R* also reduces the proteosomal degradation of *PTEN* (phosphatase and tensin homologue) [19].

*PTEN* is a tumor suppressor gene that acts as a phosphatidylinositol-3,4,5-triphosphate (PIP_3_) phosphatase, degrading PIP_3_ to PIP_2._[20]. High levels of PIP_3_ stimulate the phosphorylation and over-activation of the Akt-mTOR signaling pathway, inducing aberrant cell growth and inhibition of apoptosis [20]. As a result, in UVB-exposed cells, *WT-MC1R* protects *PTEN*’s inhibitory function over the PI3K-Akt-mTOR signaling pathway by keeping in check PIP_3_ cell levels, thus preventing aberrant cell development, whereas *MC1R* “red hair color” mutant variants are more often associated with *PTEN* inactivation. 

There are studies that identified a correlation between certain groups of gene defects and the nature and evolution of gliomas. The most noteworthy genetic hallmarks are: *IDH1/2* mutations, 1p/19q co-deletion, *MGMT* (O-Methylguanine-DNA Methyltransferase) promoter methylation status, *TP53* expression, Ki-67, *ATRX* (X-linked α-Thalassemia/mental retardation syndrome) mutation, *TERT* (telomerase reverse transcriptase) mutation, EGFR (epidermal growth factor receptor) expression, *PTEN* mutations, *CDKN2A* mutations, as well as *BRAF* mutations [21,22,23,24].

EGFR is a tyrosine kinase receptor that is involved in brain development, stimulating proliferation, migration and differentiation of all types of central nervous system cells. Secondary to ligand-receptor interaction, activated EGFR increases phosphorylation of the Ras pathway [24]. Ras is a G-protein that becomes active through the phosphorylation of GDP to GTP (Ras-GTP bound) and controls cell proliferation, survival and migration through the MAPK/ERK (mitogen-activated protein kinase) downstream signaling pathway. In addition, EGFR promotes the activation of PI3K (phosphatidylinositol-4,5-bisphophate 3-kinase) which phosphorylates the 3′ position of the inositol ring of PIP_2_, thus generating high levels of PIP_3_ [25]. In turn, PIP_3_ triggers the phosphorylation of different targets from the Akt pathway, thus providing cell signals that promote survival, growth, and proliferation [25].

Therefore, the characteristic EGFR overexpression described in glioblastomas accounts for an excessive activation of Ras-MAPK linear and PI3K-Akt non-linear pathways which are responsible for aberrant cell growth and cell apoptosis inhibition [24,25]. In addition, *PTEN* gene silencing would lead to even more uncontrolled levels of PIP_3_ due to the lack of degradation to PIP_2_. As a result, *PTEN* inactivation may represent another genetic hallmark of the melanoma–astrocytoma syndrome.

## 2. Case Presentation

We present the case of a 63-year-old male patient who was admitted to our clinic with a 1.5 cm, asymptomatic, asymmetric, dark brown-black nodular tumor, located on the right posterior hemi-thorax. (Figure 1) According to the patient’s history, it had developed in approximately three months, on the site of a former nevus. Dermoscopy revealed vascular polymorphism on an unevenly colored background (red, black, dark brown) with asymmetric borders, discretely covered by a bluish veil (Figure 2). The patient denied any personal or family history of malignancy. The clinical examination revealed two more dysplastic nevi on the anterior thorax, but no other noteworthy signs or symptoms suggestive of regional (palpable lymphadenopathy) or systemic metastatic disease (Figure 3).

We performed an excisional biopsy of the tumor and sent it for histopathological examination, which described nests of polymorphic melanocytes invading the reticular dermis (Clark level IV), with a Breslow index of 5 mm, intense mitotic activity (12 mitosis/mm^2^), chronic inflammatory infiltrate surrounding the base of the tumor, absence of ulceration and lymphovascular invasion (Figure 4). In addition, immunohistochemical staining was also performed and it was positive for S100 protein and Melan-A, as well as Ki-67 which was positive in 40% of the tumor cells (Figure 5). Therefore, correlating these results, we established the diagnosis of T4a nodular malignant melanoma—clinical stage IIB according to AJCC 8th ed. melanoma staging [26]. Conformable to the American Cancer Society, the five-year survival rate for stage IIB melanoma is 70%, while the ten-year survival rate is around 57%. 

In order to assess the presence of metastatic disease, we performed a complete ultrasound examination of the upper body main node stations, as well as thoracic, abdominal and pelvic computed tomography (CT) scans, and head magnetic resonance imaging (MRI). Because these imaging techniques did not reveal any abnormal modifications, we referred the patient for a sentinel lymph node biopsy, which was performed simultaneously with the wide local re-excision with 2 cm margins. The sentinel lymph node biopsy described no metastatic involvement of the axillary nodes (Figure 6). As a result, the patient was referred to the Oncology department with a pathological stage IIB (pT4aN0M0).

Following the European interdisciplinary melanoma guideline, the Oncology department decided to initiate a low dose adjuvant therapy with interferon-α—3 million IU s.c., three days/week, for twelve months [27]. As far as follow-up was concerned, the patient was instructed to present for a dermatological skin examination every three months. In addition, the oncologist recommended a CT scan of the thorax, abdomen, and pelvis as well as a bone scintigraphy twice per year. Moreover, cerebral MRI and tumor markers dosage (CEA, CA, S100, PSA) every three months were included in the follow-up plan for the patient.

Considering the histopathological risk factors of developing metastasis, the patient was referred to test the *BRAF* gene mutation, which revealed no mutations of the V600 position (Figure 7). 

Approximately 15 months after the primary excision, a follow-up cerebral MRI revealed a 30/20 mm left frontal lobe tumor, for which the patient was referred to the Neurosurgery department (Figure 8). The patient did not present any neurological symptoms and, due to his medical history, the first suspicion was that of a melanoma metastasis. As a result, Gamma knife therapy was recommended, but it failed to control the tumor, thus requiring direct excision.

The histopathological report described multinucleated giant cells as well as spindle and round cells with hyperchromatic nuclei and an increased nuclear to cytoplasmic ratio. In addition, numerous atypical mitoses, hyperplastic vascular structures, and necrotic foci were also present. These aspects, correlated with the rapid growth rate (less than three months), were suggestive of a diagnosis of glioblastoma multiforme (Figure 9).

Immunohistochemical staining was positive for Vimentin, S100 protein, but negative for Melan-A, which excluded a melanocytic origin. In addition, glial fibrillary acidic protein (GFAP) and isocitrate dehydrogenase-1 *(IDH1*) were diffusely positive, while Ki-67 was positive in approximately 40% of the tumor cells. Immunohistochemistry confirmed the diagnosis of high-grade glioblastoma, but its primary or secondary nature was debatable. 

On the one hand, *IDH1* staining targets the IDH1-R132H mutation characterized by the replacement of arginine with histidine at codon 132, which corresponds to the enzymatic active site [28]. This mutation is present in approximately 80–90% of secondary glioblastomas [29]. On the other hand, the patient’s age (secondary glioblastomas usually affect younger people), Ki-67 positivity in 40% of the cells and the fact that the tumor developed rapidly, in less than three months, are arguments for a primary, more aggressive glioblastoma. The most probable explanation was that the last clear brain MRI missed an intermediate grade glioma (grade II or III), which transformed in the following three months in a secondary grade IV glioblastoma. However, there are also cited cases of primary glioblastoma with *IDH1* positive immunohistochemical staining [23].

Many authors questioned whether *IDH1* mutation is a main driver of oncogenesis or merely a secondary product of defective DNA repair attempts. *IDH1* oxidizes isocitrate to α-ketoglutarate by utilizing nicotinamide adenine dinucleotide phosphate (NADP+) as a cofactor which is transformed into NADPH during the reaction. As a result, *IDH1* protects the cell from oxidative damage, but it can also reduce α-ketoglutarate to isocitrate during periods of cellular hypoxia or mitochondrial dysfunction [29].

Dang et al. discovered that the IDH1-R132H mutation of the enzymatic active site leads to a gain-of-function, *IDH1* mutants promoting an NADPH-dependent reduction of α-ketoglutarate to 2-hydroxiglutarate [30]. Moreover, in the frequent cases of IDH1-R132H codominance with wild-type *IDH1*, the latter directly generates NADPH and α-ketoglutarate which are consumed by the mutant gene copy [30]. This explains why malignant gliomas expressing *IDH1* mutations are characterized by elevated levels of 2-hydroxiglutarate, similar to patients with 2-hydroxiglutarate dehydrogenase deficiencies. In these cases, as well, elevated levels of 2-hydroxiglutarate correlate with a higher risk of central nervous system malignancies [30].

Xu et al. termed 2-hydroxiglutarate an “oncometabolite” because it occupies the same active site as α-ketoglutarate, thus inhibiting α-ketoglutarate-dependent dioxygenases which normally control histone demethylation [31]. As a result, elevated levels of 2-hydroxiglutarate interfere with histone and DNA methylation levels which directly regulate transcriptional activity [31]. Moreover, depletion of α-ketoglutarate inhibits prolylhydroxylase domain enzymes (PHD) that normally degrade the hypoxia-inducible factor 1α (HIF-1α) [28]. Therefore, in an *IDH1* mutant tumor cell, high levels of HIF-1α promote angiogenesis and tumorigenesis [31].

Although secondary glioblastomas are less frequent than primary ones, establishing the correct nature of the tumor and its genetic signature has an important prognostic value and it enables the prescription of targeted therapies, if available.

In their meta-analysis, Crespo et al. gathered the incidences of specific genetic alterations that are associated with primary, as well as secondary, glioblastoma, thus contouring a more complete genetic background, based on a larger dataset [24]. The results revealed that primary glioblastoma is possibly triggered by the correlation of: EGFR amplification, *TERT* mutation, *CDKN2A-*p16^INK4a^ deletion, *PTEN* mutation, and loss of heterozygosity 10p/10q [24]. On the other hand, secondary glioblastoma is characterized by: promoter methylation of *MGMT*, *TIMP-3*, *CDKN2A-*p14^ARF^ and *CDKN2A-*p16^INK4^, *IDH1* mutation, *p53* mutation, and loss of heterozygosity 22q/13q/19q [24].

In our case, because personalized targeted therapies were unavailable, and considering the economic factors, additional tests, such as determining the *MGMT* promoter methylation status or EGFR expression, were not performed.

The patient did not present any signs or symptoms of neurologic deficits post-surgery; therefore, his Karnofsky performance status (KPS) was calculated to 100 points, which allowed the oncologist to suggest a more aggressive treatment scheme. Following National Comprehensive Cancer Network (NCCN) guidelines, we administered a 60 Gy dose split in 2 Gy fractions over six weeks, five days/week. Once the patient completed the radiotherapy sessions, adjuvant chemotherapy was initiated with temozolomide 200 mg/m^2^, on a 5/28 cycle (five days in a row, followed by a break of twenty-eight days) [32]. The follow-up brain MRIs did not reveal any tumor activity or recurrences.

## 3. Results

Due to the fact that the patient developed two distinct primary cancers, both having neuroectodermal origins, we started analyzing the possibility of facing a melanoma-astrocytoma syndrome [2]. Therefore, we investigated the common gene defects that can trigger this syndrome and considered *CDKN2A*, situated on chromosome 9p21, as the main candidate.

The NCCN guidelines for melanoma also recommend considering mutation testing for *p16/CDKN2A* in cases with three or more primary melanomas, or different associations of cancers, such as: melanoma, pancreatic cancer and/or astrocytoma diagnosed in the same patient or in blood relatives [32].

In our case, the patient agreed to *CDKN2A* gene sequencing from the primary melanoma, but it came back negative for any mutations in the coding region. However, we could not test for other mutations in the noncoding regions of *CDKN2A* that might affect the gene transcripts. The next step would be genetic testing for other associated gene defects, such as: *CDK4*, *TERT*, *MITF*, *BAP1*, *TP53*, and *POT1*.

## 4. Discussion

Several reports have been published describing the development of different malignancies with common embryologic precursors or signaling pathways, either in the same patient or in the same family, as well as the associated gene defects.

Frigerio et al. studied two young patients who developed multiple cutaneous melanomas eight to ten years after undergoing radiotherapy for childhood astrocytomas [33]. For the female patient, array comparative genomic hybridization revealed a large 9p21.3 hemizygotic deletion affecting the *CDKN2A* and *CDKN2B* (encoding p15 protein which inhibits *CDK4/CDK6*, alongside p16^INK4A^) gene cluster which was also detected in her twin sister [33]. However, due to the fact that their parents did not carry the 9p21.3 deletion, the conclusion was that the gene defect occurred de novo as a result of paternal allele loss [33]. On the other hand, the male patient did not carry any *CDKN2A-CDKN2B*, *CDK4*, *MC1R* or *MITF* gene defects [33]. The authors concluded that, for young patients, radiotherapy for previous cancers is a more important risk factor for the development of cutaneous melanomas than the associated gene defects. For example, the twin sister who did not undergo radiotherapy did not develop cutaneous melanomas, even though she carried the 9p21.3 deletion, whereas the male patient presented congenital nevi associated melanoma 10 years after radiotherapy for astrocytoma, in the absence of associated gene defects [33]. Similar conclusions have been drawn by Braam et al. who studied 151,575 childhood cancer survivors and observed that 5.3% of all second malignant neoplasms were melanomas (212 out of 4010 patients that secondarily developed other cancers) and that the main associated risk factors were: radiotherapy, alkylating agents and anti-mitotic drugs [34].

Moreover, Pasmant et al. studied a large French family (approximately 100 members with eight individuals diagnosed with cutaneous melanoma and nine individuals affected by a variety of nervous system tumors, six of them presenting both entities) and discovered a large germline deletion, spanning over 403-kb, affecting the *CDKN2B-CDKN2A* gene cluster [35]. Furthermore, the authors additionally identified a large antisense noncoding RNA, termed *ANRIL*, whose gene overlapped the two exons of *CDKN2B* and had the 5′ end of its first exon located near the transcription start site of *p14/ARF* gene, which explained why *ANRIL* exhibited a stronger correlation for coordinated transcription with *p14/ARF* [35]. The authors hypothesized that in normal tissues noncoding RNAs may act as gene transcription regulators through different possible mechanisms, such as RNA interference, gene cosuppression, and DNA demethylation (i.e., *ANRIL* could regulate *p14/ARF*, as well as *p15/CDKN2B* and *p16/CDKN2A* expression), thus explaining why large deletions including the *ANRIL* gene could further affect tumor suppressor gene transcripts [35,36]. 

## 5. Conclusions

Though exceptionally, there are reports in the literature on the occurrence of both melanoma and glioblastoma, either in the same patient or in the same family. The genetic basis of these associations has neuroectodermal origins and was clarified by studies that detected depletion in genetic loci encoding essential tumor suppressor genes [37]. This association is now officially known as the familial melanoma–astrocytoma syndrome [37].

Even though secondary glioblastomas are less frequent than primary ones, establishing the correct nature of the tumor and its genetic signature has an important prognostic value and it enables the prescription of targeted therapies, if these are available. Studies show that survival rates from glioblastoma are poor, with a median survival of only three months in untreated patients [38]. Only a few patients survive two and a half years, and fewer than 5% survive five years following a glioblastoma diagnosis [39]. 

Despite current studies, an evident syndrome describing the association between melanoma and glioblastoma has yet to be determined. From our point of view, the presented case report describes a patient with a history of stage IIB malignant melanoma and primary glioblastoma. This unique case report illustrates both histologic and genetic diversity and raises awareness regarding the sporadic co-occurrence of these two medical entities. We believe that the co-occurrence of melanoma and glioblastoma is likely not coincidental. Therefore, more insight and research are needed to fully understand this unique association. 

## Figures and Tables

**Figure 1 diagnostics-13-01046-f001:**
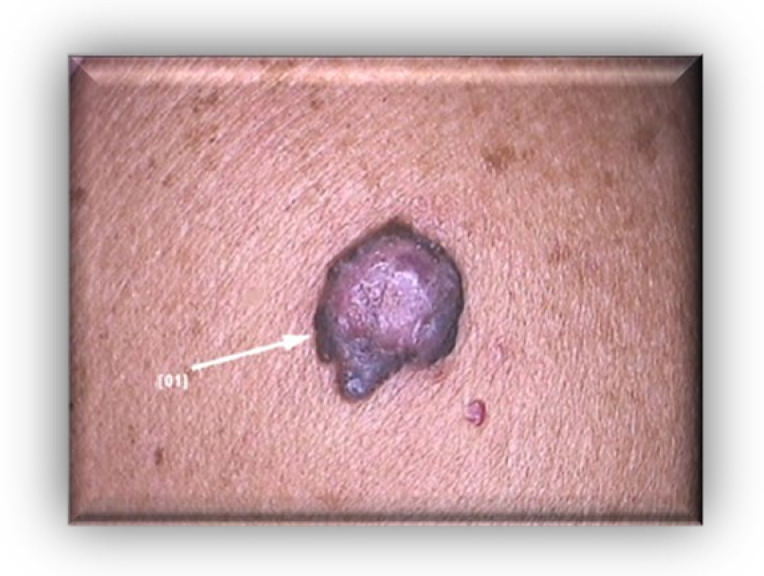
Clinical aspect: asymmetric, dark brown-black nodular tumor, located on the right posterior hemi-thorax.

**Figure 2 diagnostics-13-01046-f002:**
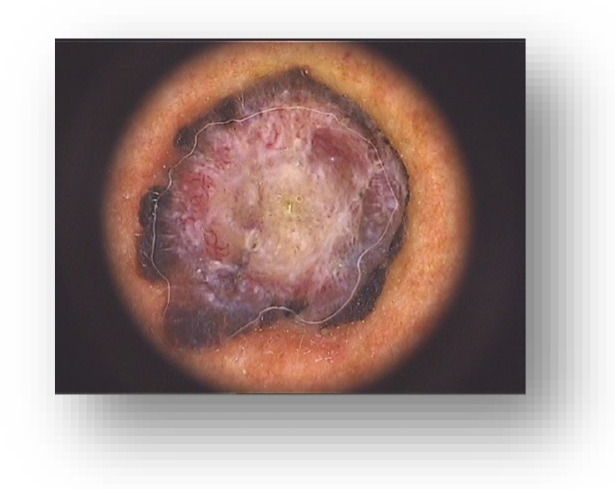
Dermoscopy findings: vascular polymorphism on an unevenly colored background (red, black, dark brown) with asymmetric borders, discretely covered by a bluish veil.

**Figure 3 diagnostics-13-01046-f003:**
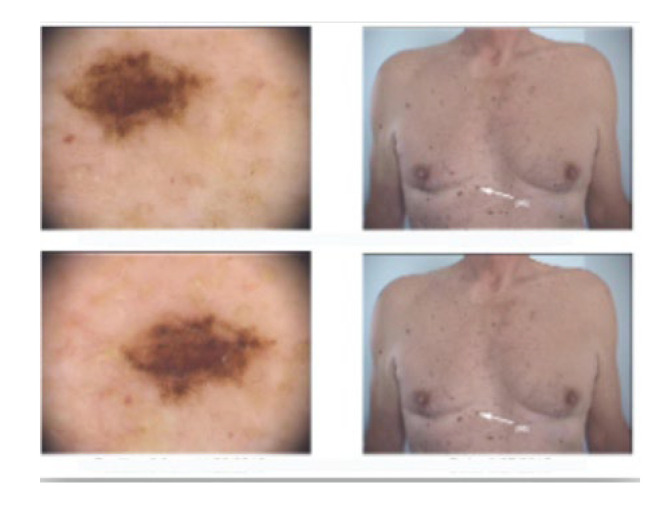
Clinical and dermoscopical images of two more dysplastic nevi on the anterior thorax.

**Figure 4 diagnostics-13-01046-f004:**
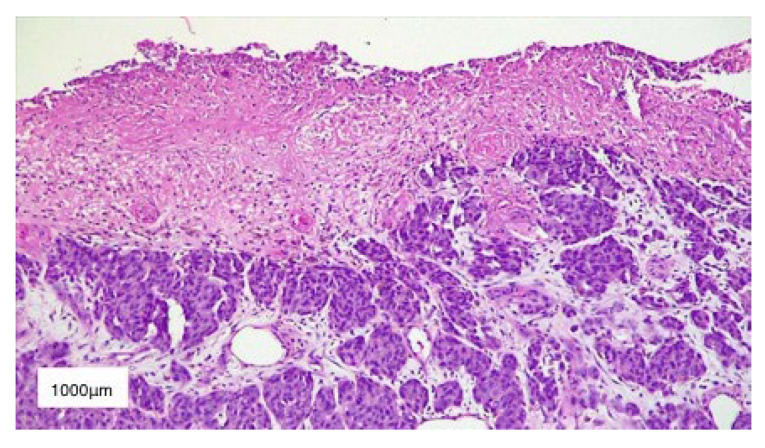
HE-stained (10×) histopathological section of Clark IV malignant melanoma. Scale bar: 1000 μm.

**Figure 5 diagnostics-13-01046-f005:**
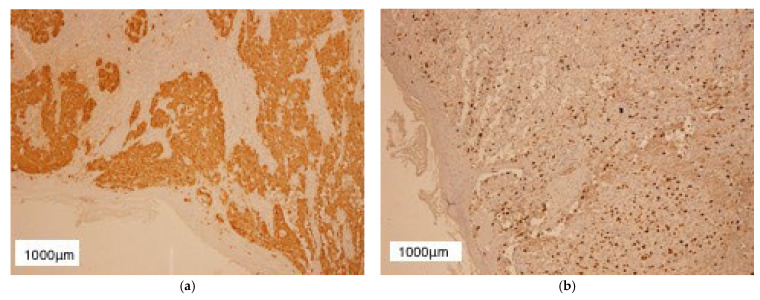
Microphotographs of malignant melanoma sections in our case. (**a**) Immunohistochemical stain (10×) positive for S100; (**b**) Immunohistochemical stain (10×) positive for Ki-67; (**c**) Immunohistochemical stain (10×) positive for Melan-A. Scale bars (**a**–**c**) images: 1000 μm.

**Figure 6 diagnostics-13-01046-f006:**
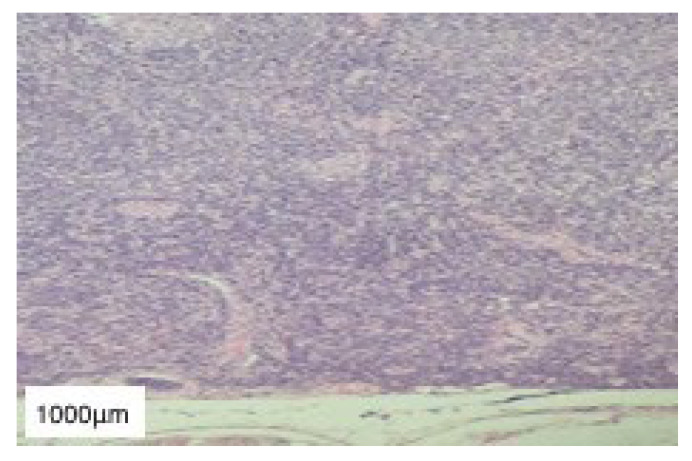
HE staining (20×) of sentinel lymph node biopsy which describes no metastatic involvement of the axillary nodes. Scale bar: 1000 μm.

**Figure 7 diagnostics-13-01046-f007:**
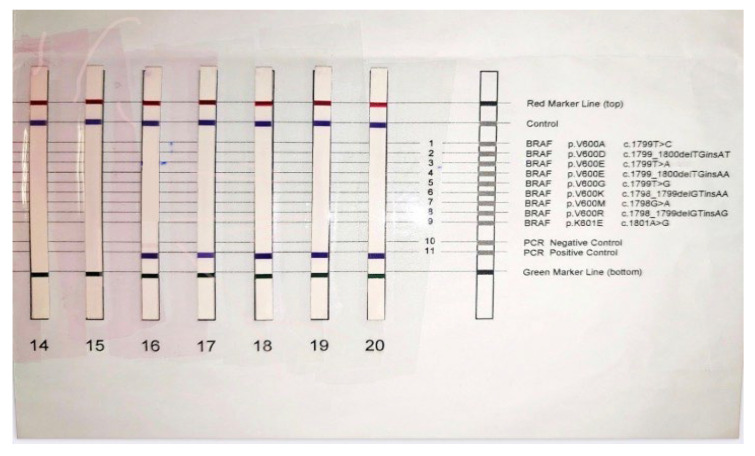
Negative BRAF gene mutation result. The genotyping was performed using the BRAF 600/601 StripAssay kit, which detects the mutations: V600E, V600A, V600D, V600E (c. 799T > A, c. 1799_1800TG > AA), V600G, V600K, V600M, V600R, K601E in the BRAF gene. The detection limit of the method is 1%.

**Figure 8 diagnostics-13-01046-f008:**
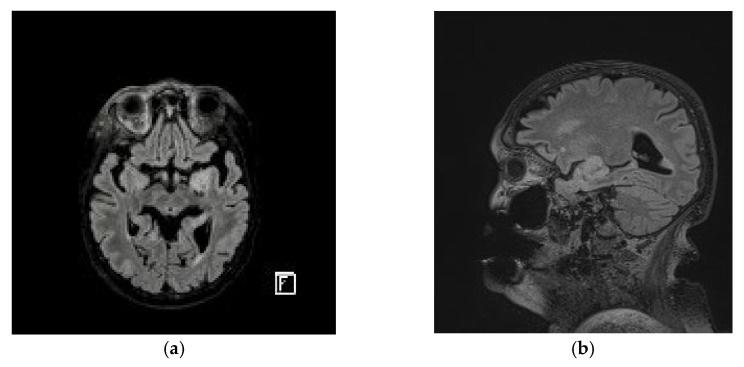
(**a**) Axial MRI image of a large heterogeneous mass in the left frontal lobe. (**b**) Lateral MRI image of a large heterogeneous mass in the left frontal lobe.

**Figure 9 diagnostics-13-01046-f009:**
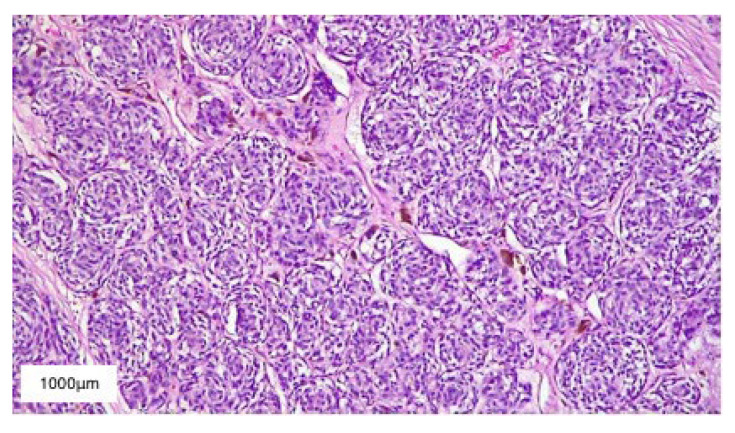
HE-stained (10×) histopathological section of glioblastoma multiforme. Scale bar: 1000 μm.

## Data Availability

Not applicable.

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
