# Peer review of "Cutaneous Melanoma and Glioblastoma Multiforme Association—Case Presentation and Literature Review"

_diagnostics, 2023, doi:10.3390/diagnostics13061046_

Round 1

Reviewer 1 Report

Dear Authors,

I have already reviewed the previous version of this manuscript, when you submitted it in Life (Section: Physiology and Pathology; Special Issue: Current Research on Dermatology: Pathology, Clinical Manifestation, Investigation and Therapy). I believe that the manuscript, now submitted in Diagnostics, has not been significantly improved in term of data presentation and does not warrant publication in Diagnostics in the current state. Further major revisions are required.

1. Compared to the previous version submitted in Life, the manuscript has been improved only in term of a Hematoxylin/Eosin (H&E) staining. The results of immunohistochemical analyses have been well described in the text, but as previously requested, it is necessary to show immunohistochemical staining for typical markers of skin tumor, as well as Vimentin, S100, Melan-A and Ki-67. The fact that the patient died due to other circumstances does not represent a right motivation for not provide any additional analyses besides these new ones, since this involves using the same paraffin-embedded samples used for the H&E staining. Similarly, for the ultrasound imaging analyses (CT scans and MRI) of the main body of tumor, and the sentinel lymph node biopsy. 

2. For the determination of the presence of BRAF gene mutation, the authors did not use Next Generation Sequencing (NGS) technology, but a Real-Time PCR-based assay for V600 mutations in the BRAF gene. However, no data are provided. The reviewer understands that NGS technology can be not available at the time and too expensive, but it would be at least appropriate: a) to show the Real-Time PCR data and b) to perform a DNA sequencing of the specific BRAF gene mutation region and show the results of the electropherogram. 

3) H&E stainings need to be improved: scale bars are missing and no information is specified regarding the magnification (x) and the length (μm) in the Figure Legend.

In light of the above, the manuscript is still preliminary and the quality of the data does not reach the expected quality standards required for publication. 

Author Response

Esteemed reviewer, thank you very much for your valuable suggestions. Here are the changes that have been made accordingly to your requests.

Point 1:  Compared to the previous version submitted in Life, the manuscript has been improved only in term of a Hematoxylin/Eosin (H&E) staining. The results of immunohistochemical analyses have been well described in the text, but as previously requested, it is necessary to show immunohistochemical staining for typical markers of skin tumor, as well as Vimentin, S100, Melan-A and Ki-67. The fact that the patient died due to other circumstances does not represent a right motivation for not provide any additional analyses besides these new ones, since this involves using the same paraffin-embedded samples used for the H&E staining. Similarly, for the ultrasound imaging analyses (CT scans and MRI) of the main body of tumor, and the sentinel lymph node biopsy. 

Response 1: The text has been modified accordingly.

Point 2:  For the determination of the presence of BRAF gene mutation, the authors did not use Next Generation Sequencing (NGS) technology, but a Real-Time PCR-based assay for V600 mutations in the BRAF gene. However, no data are provided. The reviewer understands that NGS technology can be not available at the time and too expensive, but it would be at least appropriate: a) to show the Real-Time PCR data and b) to perform a DNA sequencing of the specific BRAF gene mutation region and show the results of the electropherogram. 

Response 2: Since the patient did not show any mutation of the V600 position, we do not consider of clinical interest to attach the result.

Point 3:  H&E stainings need to be improved: scale bars are missing and no information is specified regarding the magnification (x) and the length (μm) in the Figure Legend.

Response 3: The text has been modified accordingly.

Reviewer 2 Report

In the manuscript” Cutaneous melanoma and glioblastoma multiforme association – case presentation and literature review” Orzan and colleagues have presented a case report of a 63-year-old patient who developed glioblastoma after undergoing surgery and treatment for melanoma. Although the manuscript describes a single case report, is well structured and will advance with understanding in this field of melanoma-astrocytoma syndrome.  However, some issues should be addressed before acceptance for publication.

Major:

1.In the manuscript authors comments results of immunohistochemical staining for vimentin, S100, Melan-A, Ki-67, however these results are not included in the manuscript, representative images should be shown also in this manuscript.  

2. Please make sure to follow the rules of writing genes (italic) vs proteins names (regular text) when referring to their names in the main text, everything is italic and it is confusing. For example, the markers for IHC analysis are proteins which have been stained for (vimentin, S100, Melan-A), signaling pathways (MAPK/ERK, AKT-TOR and so on) are all proteins and should not be in italic

3. Some English fluency and reading accessibility to the non-specialist readers need revision.

Author Response

Esteemed reviewer, thank you very much for your valuable suggestions. Here are the changes that have been made accordingly to your requests.

Point 1:  In the manuscript authors comments results of immunohistochemical staining for vimentin, S100, Melan-A, Ki-67, however these results are not included in the manuscript, representative images should be shown also in this manuscript.  

Response 1 The text has been modified accordingly.

Point 2 Please make sure to follow the rules of writing genes (italic) vs proteins names (regular text) when referring to their names in the main text, everything is italic and it is confusing. For example, the markers for IHC analysis are proteins which have been stained for (vimentin, S100, Melan-A), signaling pathways (MAPK/ERK, AKT-TOR and so on) are all proteins and should not be in italic

Response 2: The text has been modified accordingly.

Point 3: Some English fluency and reading accessibility to the non-specialist readers need revision.

Response 3: The text has been modified accordingly.

Round 2

Reviewer 1 Report

1. H&E and immunohistochemical stainings need to be improved: although the authors specified information regarding the magnification (x), scale bars in the figures are still missing and no informations are specified regarding the length (μm) in the Figure Legend.

2. Although the patient did not show any mutation of the V600 position, the reviewer retains that it is appropriate to attach the results obtained since the authors referred this data in the manuscript. 

Round 3

Reviewer 1 Report

Scale bars in the figures are still missing. Please, add them.

Thereafter, the paper is ready for the publication in Diagnostics.
